# Peripheral Bone Relapse of Paediatric TCF3-HLF Positive Acute Lymphoblastic Leukaemia during Haematopoietic Stem Cell Transplantation: A Case Report

**DOI:** 10.3390/children9121919

**Published:** 2022-12-08

**Authors:** Máté Horváth, Gabriella Kertész, Csaba Kassa, Vera Goda, Kata Csordás, Lidia Hau, Anna Kövér, Anita Stréhn, Orsolya Horváth, Krisztián Kállay, Gergely Kriván

**Affiliations:** 1Károly Rácz Doctoral School of Clinical Medicine, Semmelweis University, H-1097 Budapest, Hungary; 2Pediatric Hematology and Stem Cell Transplantation Unit, Central Hospital of Southern Pest National Institute of Hematology and Infectious Diseases, Albert Flórián Street 5-7, H-1097 Budapest, Hungary

**Keywords:** acute lymphoblastic leukaemia, haematopoietic stem cell transplantation, osteomyelitis, relapse, extramedullary relapse, bone relapse, TCF3-HLF

## Abstract

The present case report features a highly uncommon form of a paediatric TCF3-HLF positive acute lymphoblastic leukaemia (ALL) relapse, an extramedullary, peripheral bone manifestation. Following complete remission, during the conditioning for haematopoietic stem cell transplantation (HSCT), our sixteen-year-old male patient complained of fever, pain and swelling of the right forearm. Radiography suggested acute osteomyelitis in the right ulna with subsequent surgical confirmation. Intraoperatively obtained debris culture grew *Staphylococcus aureus* and *Acinetobacter pittii*. Measures taken to control the infection were deemed to be successful. However, after the completion of the otherwise uneventful HSCT, a very early medullary relapse was diagnosed. Revising the original surgical samples from the ulna, bone relapse of ALL was immunohistochemically confirmed. Reviewing the previous cases found in the literature, it is advised to consider uncommon forms of ALL relapse when encountering ambiguous cases of osteomyelitis or arthritis during haematological remission.

## 1. Introduction

Advances in the treatment of the most common paediatric malignancy, acute lymphoblastic leukaemia (ALL), can be undoubtedly regarded as a success story. Even though five-year overall survival rates are >85%, there is still a need for improvement in high-risk and relapsed subgroups of ALL [1,2,3]. Based on the immunophenotype and the cytogenetic profile of malignant cells, the patient’s age, the initial response to the therapy, and the involvement of extramedullary sites, high-risk subgroups are defined and associated with inferior outcomes. Approximately 25–30% relapse rate can be observed in these groups [3]. This strikes a note of caution, for the second most common malignancy-associated death in children is the relapsed ALL.

The majority of ALL relapse occurs within 2–2.5 years after the initial diagnosis, primarily in the bone marrow, followed by ‘conventional’ extramedullary locations, such as the central nervous system and, for male patients, the testis [4,5]. However, there are a handful of reports about unusual locations for leukaemic infiltrates, such as the kidney, skin, eyes, ovary, and bones [6]. The rarity and the atypical presentation of such isolated relapses with apparent bone marrow remission pose a diagnostic challenge for clinicians. 

Isolated extramedullary relapse of leukaemia in the skeletal system, though an extremely rare phenomenon, is a matter of greater concern. There is a limited number of case reports regarding this topic [6,7,8,9,10,11,12,13,14,15]. In 1994, Murray et al. reviewed the available reports at that time and made several important observations [7]. These isolated bone relapses occurred during haematologic remission, the majority of them several years after the initial diagnosis. A significant portion of the patients had suffered bone marrow relapse or extramedullary relapse of more ‘conventional’ anatomical sites (central nervous system, testis) beforehand. Moreover, the isolated bone involvement had a high chance of progressing into definitive medullary disease. Despite aggressive therapeutical measures, all but one case led to consequent death. The underlying mechanism of extramedullary proliferation perplexes clinicians. The road to definitive diagnosis is seldom paved without ambiguity. The initial symptoms and non-specific radiographic findings can imitate infectious or immune-mediated pathologic entities of the musculoskeletal system [16]. Normal complete blood count (CBC) is not unprecedented either [17]. This equivocality may contribute to the delay of diagnosis. As a high-risk form of leukaemia is portrayed by all these aggressive features, such delay is not without its consequences.

## 2. Case Presentation

### 2.1. Diagnosis and Initial Treatment

A sixteen-year-old adolescent male diagnosed with ALL was admitted to our transplantation unit for allogeneic haematopoietic stem cell transplantation (HSCT). Initial signs of the disease had been caught 8 months earlier. His treatment had been commenced based on the ALL IC-BFM 2009 study protocol of the International Berlin–Frankfurt–Münster Study Group [18]. He had been provisionally assigned to the intermediate risk group. On the 15th day of remission induction, bone marrow aspiration from the iliac crest revealed a high rate of residual lymphoblasts (14%). Moreover, RNA sequencing (Illumina Trusight RNA Pan-Cancer Panel) had identified the TCF3-HLF fusion gene, prognosticating an extremely poor outcome. Consequently, the therapy was escalated to the high-risk group with the plan of allogeneic HSCT after the consolidation. Further molecular genetic analysis of the lymphoblast genome (QIAseq targeted DNA panel; SALSA MLPA) revealed several pathogenic mutations, some of which conferred an unfavourable outcome or a high relapse rate (the genes concerned: KRAS, ZEB2, CDKN2A, and B). Morphologic remission had been achieved after 33 days. The remaining therapy had been completed successfully, without any unmanageable complications and confirmed negative, flow cytometric minimal residual disease (MRD) 3 months before the planned HSCT. MRD assessment targeting the pathogenic TCF3-HLF fusion had not been available for us. Cerebrospinal fluid involvement of leukaemia was not detected during his treatment.

### 2.2. The Course of HSCT

In view of the patient’s high-risk leukaemia, the following treatment was decided for conditioning: total body irradiation (TBI) with 12 gray total doses (delivered in four fractions on pre-transplant days 8–4), combined with etoposide-phosphate (60 mg/kg on pre-transplant day 2) and anti-thymocyte globulin (15 mg/kg between pre-transplant days 4–1) [19]. At the start of the conditioning, white blood cell and neutrophil cell counts were in the normal range. His anti-infectious prophylactic regimen included levofloxacin, acyclovir, and posaconazole. Shortly before the planned TBI, the patient had become febrile and empiric piperacillin/tazobactam had been initiated. Considering the patient’s overall stable state, with no markedly increased inflammatory parameters, the decision was made for commencing the irradiation therapy. The patient remained afebrile during the first days of the irradiation. Later, however, a marked surge of CRP and procalcitonin was observed. Moreover, significant pain and swelling of the right forearm developed. As the procalcitonin surge correlated with the irradiation, the possibility of malignancy was not dismissed. Radiography of the right forearm proposed extensive osteomyelitis based on the thinned cortex, a pattern of cortical destruction and a periosteal reaction (Figure 1). Upon getting new clues for a presumed infection and consultation with orthopaedic surgeons, the patient’s anti-microbial treatment was changed to meropenem—vancomycin combination, and he was immediately admitted to an orthopaedic/trauma unit after the completion of the TBI. Surgical irrigation and thorough debridement were performed on the right ulna, which presented the picture of extensive, purulent-necrotic osteomyelitis. Gentamicin-containing polymethyl methacrylate beads were placed into the evacuated marrow. The radius of the same forearm remained unaffected. 

After the successful operation, the conditioning protocol was continued with no further obstruction. Upon completion, a peripheral stem cell graft with adequate CD34+ and T-cell count was administered from an 11/12 human leukocyte antigen-matched unrelated donor. For the prevention of graft versus host disease, the patient received cyclosporine A and methotrexate. 

During the pre-engraftment phase, recurring fever was observed with stable haemodynamic parameters and without any obvious causative pathogen from routine blood cultures. Intraoperatively obtained pus culture, on the other hand, after several days of incubation, grew methicillin-sensitive Staphylococcus aureus and Acinetobacter pittii. The histopathological analysis did not suggest malignancy at that time. Based on further antimicrobial sensitivity tests, targeted osteomyelitis treatment was carried out with cefepime (from day +7) and ciprofloxacin (from day +13). In parallel, new onset pain in the left forearm prompted fear of osteomyelitis affecting other parts of the skeletal system. Ultrasound imaging presented this possibility for the left ulna. Consultation with surgeons deemed the state of the left forearm unnecessary for immediate invasive intervention, and the intravenous antimicrobial treatment was carried on unaltered. 

On day +17, increasing leukocyte and neutrophil granulocyte counts indicated engraftment. Three weeks after the transplantation, the patient presented with improving and stable haematopoiesis without any clinical signs of contiguous extension or systemic progression of the osteomyelitis. From day +28, cefepime was omitted from the targeted anti-microbial treatment, and trimethoprim/sulfamethoxazole and ciprofloxacin combination was used for the long-term maintenance therapy. Complete donor chimerism (100%) was achieved according to the short tandem repeat analysis. Iliac crest bone marrow aspiration was performed on day +26 for post-transplantation MRD assessment. Flow cytometric analysis of the marrow revealed a small population of immature B-cells (0.5%) with some phenotypic characteristics atypical for normal B-cell development. As the presence of the malignant B-cell clone could not be verified unambiguously, close and regular follow-up was planned.

### 2.3. Early Relapse and the Final Course of the Disease

Further chimerism assays indicated the stable presence of donor cells. Routine complete blood tests were normal, but a trend of increasing white blood cell and absolute lymphocyte count could be observed. On day +47, the white blood cell count rose considerably high, and immediate iliac crest bone marrow aspiration was performed: early relapse was diagnosed with the presence of malignant B-cells (53%) in the marrow. His treatment was continued according to the ALL IC REL 2016 HI protocol [20]. Within a few days, pancytopaenia developed, along with severely impaired renal function, and the patient had to be transferred to the intensive care unit. In the following weeks, overall health progressively deteriorated with multifocal bacterial infections and adenovirus encephalitis. The patient passed 80 days after the HSCT. 

Taking into account the rapid relapse after HSCT, discussions were made about the possible origin of the presumed osteomyelitis. One of our hypotheses was that malignant lymphoblasts may had already been present in the destroyed tissues of the right ulna. This train of thought had been corroborated with the revised immunohistochemical (IHC) analysis of the bone debridement taken during the operation: cells with characteristics undoubtedly of lymphoblasts were present in bone tissue (Figure 2). For the IHC, only B-cell markers were used, as the samples had partially deteriorated. The final conclusion can be drawn as by the time the preparation for HSCT was initiated, the patient had already had extramedullary bone relapse manifesting in the clinical picture of osteomyelitis. Routine chest radiography and computed tomography before the HSCT did not indicate malignancy in the axial skeleton. However, without positron emission tomography, the full extent of the relapse cannot be known. The main events of the case are shown on Figure 3.

## 3. Discussion

We report a case of an extremely rare and early occurrence of extramedullary ALL relapse, a peripheral bone involvement. Reviewing the literature, only a handful of reports are available regarding this topic [6,7,8,9,10,11,12,13,14,15]. In 12 out of the 15 paediatric cases featured, subsequent bone marrow or central nervous system relapse was observed, with eventual death in seven of these. The median time from the diagnosis of the original leukaemia to the bone relapse was 6 years, and several patients (eight) had already experienced a bone marrow relapse beforehand. The most affected parts of the skeletal system were the lower extremities (7) and the mandible (5). However, most of these reports were made before 2000. Therefore, caution must be applied, as these observations cannot be fully extrapolated to present cases. Our case is unique in that the bone relapse occurred early (just 3 months had passed between the last bone marrow aspiration and the osteomyelitis surgery).

The most relevant finding of the molecular genetic analysis of the patient’s lymphoblasts was the TCF3-HLF fusion. This gene fusion describes a rare sub-type of paediatric ALL with an exceptionally poor outcome and high relapse rate even with HSCT [21]. A recent report by Takahashi et al. [15] describes an isolated relapse of TCF3-HLF positive ALL after allogeneic HSCT, with multiple skeletal involvement. In the literature, it has been postulated that the TCF3-HLF might induce the production of parathyroid hormone-related peptide (PTHrP) and sushi-repeat protein upregulated in leukaemia (SPRUL) as one of their downstream targets, resulting in hypercalcaemia, one of the signature traits of this sub-type [22,23]. In addition, the overexpression of SPRUL may play a role in bone invasion as an adhesion molecule [23]. Our case report, along with the aforementioned reports, stands as a further example of the bone affinity of TCF3-HLF-positive leukaemia.

As had been described before, the nonspecific clinical presentation of leukaemic musculoskeletal involvement can be attributed to infectious or immune-mediated disease of bones and joints [16]. Being in the state of complete remission with confirmed negative MRD or presenting with normal CBC steer away from the early suspicion of malignancy. Evaluation of other laboratory parameters may indicate elevated CRP or erythrocyte sedimentation rate, and in most cases, radiographic imaging uncovers the destruction and inflammation of bones and joints. However, the mentioned methods are unable to differentiate between malignancy and infection, given the nonspecific nature of these findings [16,24,25]. Recipients of HSCT are at increased risk for either common or opportunistic infections, taking into account severe neutropenia, prolonged immunodeficiency, and immune suppressive therapies [24]. In view of the latter, our diagnostic sequelae with elevated inflammatory laboratory parameters and striking radiographic findings suggested the possibility of acute osteomyelitis. The necrotic, purulent sample taken during the surgery grew Staphylococcus aureus and Acinetobacter pitti, with the former being the most commonly isolated pathogen of osteomyelitis in the paediatric population [25]. Given the presenting nature of the affected area and the detected pathogens, it is not unsubstantiated to think that the patient’s relapse was indeed complicated by acute osteomyelitis, an acknowledged but quite rare phenomenon [26]. Considering the negative blood cultures, contradiction does not necessarily arise, as causative pathogens are identified in less than 50% of the cases via this method [25]. Contamination of the samples cannot be definitely ruled out either.

To complicate the case further, ultrasound imaging suggested a similar lesion in the left forearm. This presents the possibility of a “patchy” relapse with multiple peripheral bone invasions. It is highly uncommon for ALL to present with this nature; nevertheless, there is a known example in the literature for isolated bone relapse with multifocal skeletal invasions [14,15]. To aid differential diagnosis in a case such as this, both Hangai [14] and Takahasi et al. [15] suggest the use of FDG-PET, which helps to identify malignant lesions and performing a CT-guided biopsy could establish the final diagnosis. In our case, because of the rapid progression of the relapse, such measures could not be taken. Therefore, the multiplicity of the leukaemic bone invasion cannot be unequivocally stated.

This study has potential limitations. The patient’s preceding MRD was assessed using multi-parametric flow cytometry analysis. The sensitivity of this approach is between 10^−3^–10^−4^ cells. However, molecular analysis of the fusion transcript would have been more adequate, as PCR-based methods can reach greater sensitivity (up to 10^−5^ cells) [27]. Unfortunately, this method was not available to us. Consequently, it is not unfounded that the patient had false negative flow-MRD before the conditioning. Furthermore, the correlation between the immunophenotype of the original and bone-invading blasts could not have been analysed due to the condition of the samples.

## 4. Conclusions

In conclusion, bone relapse of paediatric ALL is quite rare but not unprecedented. The nonspecific clinical presentation of such relapse, as it mimics immunologic or infectious diseases of the musculoskeletal system, poses a challenge for the differential diagnosis. Overviewing the literature, most of these cases, such as ours, feature an aggressive form of ALL with multiple relapses and unfavourable outcomes. The case presented here, which is the earliest known example of bone relapse, highlights the importance of suspicion even for the uncommon presentation of leukaemia when encountering musculoskeletal pathologic entity in the remission of ALL. Functional imaging techniques, such as PET, biopsy, or thorough histopathological analysis of surgical samples may be of aid in uncovering unprecedented, ambiguous cases. 

## Figures and Tables

**Figure 1 children-09-01919-f001:**
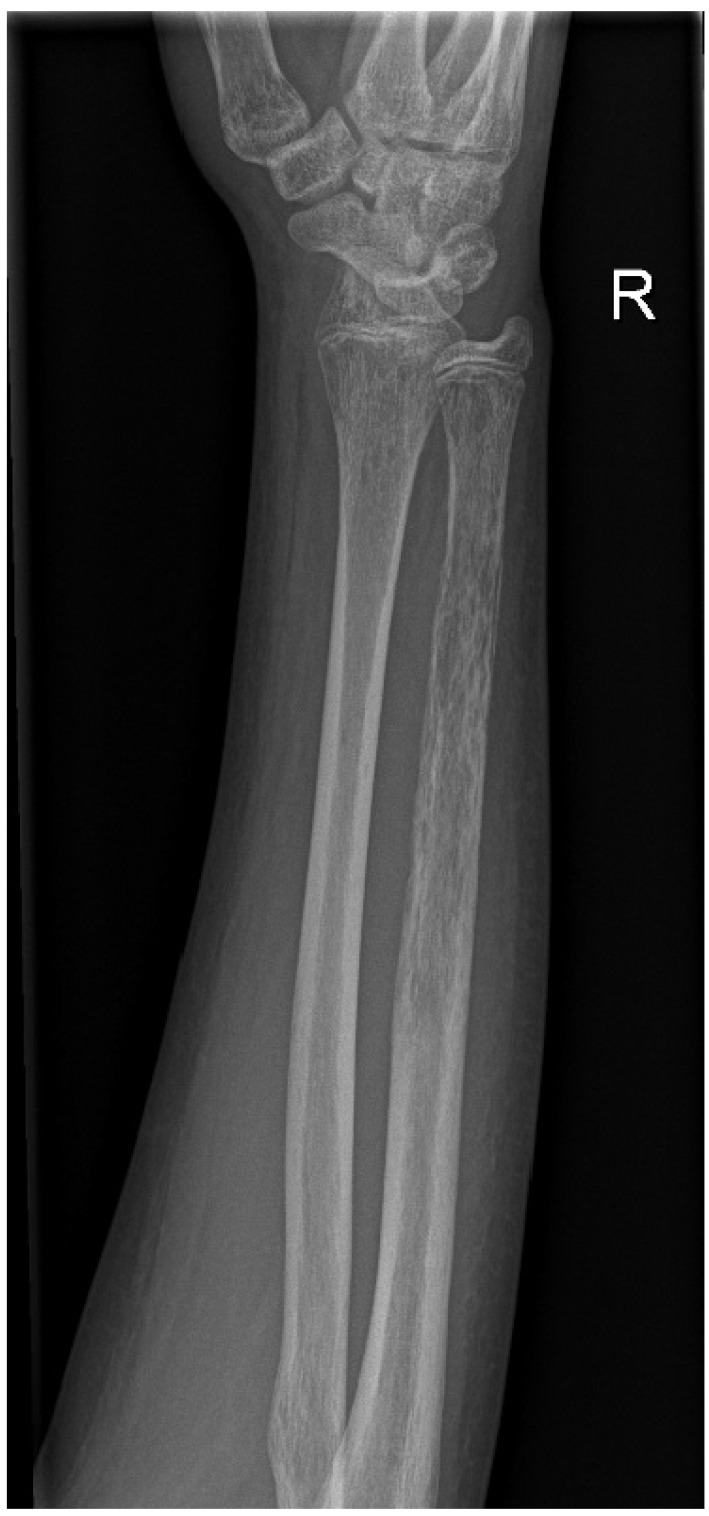
Radiography of the right forearm: thinned cortex, a pattern of cortical destruction and periosteal reaction of the ulna.

**Figure 2 children-09-01919-f002:**
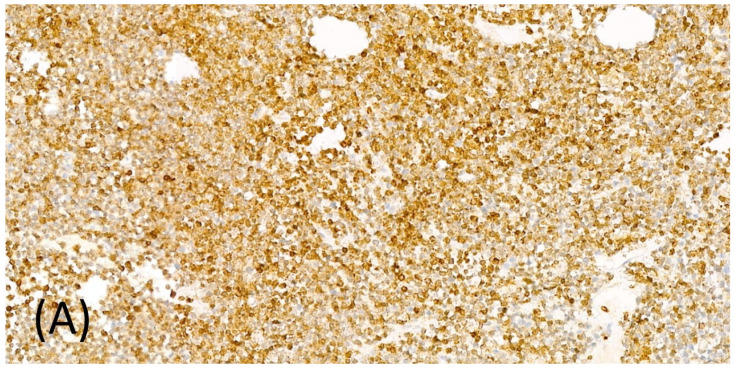
Immunohistochemistry of the original surgical samples taken from the right ulna, with CD79a (**A**) and CD10 (**B**), shows the presence of lymphoblasts in the destroyed bone tissues.

**Figure 3 children-09-01919-f003:**
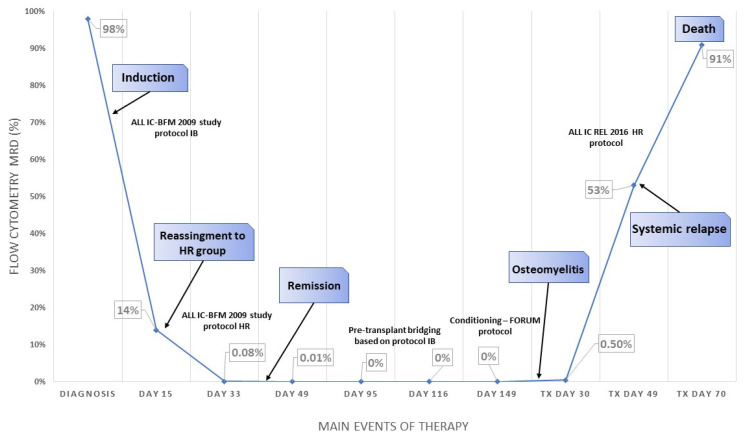
Main events of the case and flow cytometry minimal residual disease levels throughout the therapy.

## Data Availability

Not applicable.

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
