# Peer review of "Peripheral Bone Relapse of Paediatric TCF3-HLF Positive Acute Lymphoblastic Leukaemia during Haematopoietic Stem Cell Transplantation: A Case Report"

_children, 2022, doi:10.3390/children9121919_

Round 1

Reviewer 1 Report

I want to congratulate the authors for the chosen topic. It is a very interesting case, both in terms of evolution and management.  I consider it would be a gain for the article if some details about the involvement of the molecular biology (found at the case)  were added in the discussion part.

Author Response

Dear Reviewer,

The authors would like to thank you for your effort and time in reviewing our manuscript. We appreciate your kind words.

We took into consideration your comments about the underlying molecular biology and the discussion part has been updated with a small section about the fusion gene TCF3-HLF and its potential role in the skeletal invasion of leukaemia.

Máté Horváth

Reviewer 2 Report

The Authors present a case report of highly uncommon form of pediatric ALL relapse to an extramedullary, peripheral bone manifestation. However, the delay in diagnosing the relapse before transplantation depended from innacurate analysis of bone marrow and the lack of adequate imaging tests.

Comments  for the Authors:

1. Abstract: adequate

2. Introduction:  Numerous autopsy series confirmed.....

This part of the introduction is a little bit confused. If 80% of patients have leukemia in extramedullary sites without any subsequent clinical manifestations, it not clear why it is necessary to investigate for avoiding the delay of diagnosis.

3. Presentation: Diagnosis and initial treatment...

The data regarding molecular genetic analysis of the lymphoblast genome should be reported at presentation in order to outline an adequate risk profile. Based on this, the therapeutic strategy, such as allogeneic HSCT, should have been drawn.

The final conclusion....

The analysis of bone marrow before transplantation was not accurate enough and other analyses  were not  done sufficiently.

4. Discussion: our case report features...

The phrase is not clear and needs to be reformulated.

Author Response

Dear Reviewer,

The authors would like to express their appreciation for your time and effort in reviewing the manuscript.

Comment for the Introduction:

The cited sentence was written to express the peculiar nature of the presence of lymphoblasts in uncommon extramedullary sites. However, the necessity of this line can be debated and we decided to omit it altogether from the manuscript.

Comments for the Presentation:

a) This manuscript was written with the main focus on the challenges in the diagnosis of an uncommon form of leukaemia relapse that presents itself with the clinical picture of osteomyelitis. However, the TCF3-HLF fusion is a major finding in our case. Your comments about the molecular genetic background were taken into consideration. The title, the abstract, and the case presentation have been updated the reflect this. The discussion part has been updated with the potential pathogenetic role of TC3-HLF in the skeletal invasion of leukaemia.

b) The authors agree that the MRD assessment would have been more adequate if TCF3-HLF targeted PCR was used. However, this method was not available to us. This limitation must be acknowledged to understand the whole picture and is included in the final part of the discussion.

c) On a final note, we would like to respectfully share our thoughts on your comments about the inadequacy of the patient's management.

  • The plan for HSCT was decided on the grounds of the initial therapeutic response and the TCF3-HLF mutation, which confer exceptionally dire outcome and a high probability of relapse. The text has been updated to reflect on that.
  • To establish better outcome for TCF3-HLF positive patients, the efficacy of pre-HSCT blinatumomab, CAR-T or haplo-HSCT are currently underway. There is no uniformly accepted approach for these patients at present.
  • It must be noted that PCR-based MRD assessment would have found leukaemic cells in the range of 10-5 before the HSCT. However, even with this important observation, the patient conditioning would have commenced as the omission of HSCT could not have provided with the hope of remission.
  • The initial symptoms for the bone relapse presented concurrently with the conditioning. The surgery had taken place after the patient had received 12 Gy total dose of irradiation. It was not possible to perform PET at the time of the neutropenic period. Moreover, several aspects of the presentation of the bone relapse indicated infectious osteomyelitis. It is not unsubstantiated, that concurrent infection and leukaemic invasion had taken place. It can be debated that the knowledge of PCR MRD positivity would have resulted in better guidance for such an ambiguous case. Later on, the potential multiplicity of the bone invasion was indeed considered. However, the extremely rapid relapse after the HSCT hindered these efforts of ours.

In view of the aforementioned points, we would like to respectfully convey our dissent with your final conclusion.

Comment for the Discussion:

The cited phrase has been rewritten for better clarity.

Máté Horváth

Reviewer 3 Report

The limitations of the manuscript must be acknowledged by the authors.

ALL with TCF3-HLF fusion is notorious for relapse or refractoriness, even after HSCT or CAR-T. There is nothing surprising the patrient relapse. Previous doubious bone relapse is not necessary to explain that.

It is difficult to talk about ALL blasts negativity without PCR-MRD. The patient might have well presented significant PCR-MRD before transplant. Actually, TCF3-HLF patients do not achieve megative PCR-MRD.

Full immunophenotype should be presented from first diagnosis and more full correlation with immunophenotype of the IHC study of the bone.

Genetics should be presented with more detail (with method). TCF3-HLF is a very significant finding.

Full therapy course with FC-MRD timepoints should be shown.

So the paper has several limitations and the investigations are not modern. This must be acknowledged and improved before re-submission. 

Author Response

Dear reviewer,

The authors would like to express their appreciation for your time and effort in reviewing our manuscript. You made several insightful comments that made us improve the case report. We hope that you find your suggestions sufficiently incorporated into the text. These changes are highlighted within the manuscript for more clarity. Please, read below our point-by-point response to your concerns.

Concerns regarding the genetic analysis and TCF3-HLF:

This manuscript was written with the main focus on the challenges in the diagnosis of an uncommon form of leukaemia relapse that presents itself with the clinical picture of osteomyelitis. However, the TCF3-HLF fusion is indeed a major finding in our case. Your comments about the molecular genetic background were taken into consideration. The title, the abstract, and the case presentation have been updated the reflect this. The used molecular genetic methods have been included. The discussion part has been updated with the potential pathogenetic role of TC3-HLF in the skeletal invasion of leukaemia.

Concerns regarding the pre-HSCT MRD:

The authors agree that the MRD assessment would have been more adequate if TCF3-HLF targeted PCR was used. However, this method was not available to us. This limitation must be indeed acknowledged to understand the whole picture and is included in the final part of the discussion.

To better understand our position at that time, please allow us to express our thoughts on this matter.

  • The plan for HSCT was decided on the grounds of the initial therapeutic response and the TCF3-HLF mutation, which confer exceptionally dire outcome and a high probability of relapse. The text has been updated to reflect on that.
  • To establish better outcome for TCF3-HLF positive patients, the efficacy of pre-HSCT blinatumomab, CAR-T or haplo-HSCT are currently underway. There is no uniformly accepted approach for these patients at present.
  • It must be noted that PCR-based MRD assessment would have found leukaemic cells in the range of 10-5 before the HSCT. However, even with this important observation, the patient conditioning would have commenced as the omission of HSCT could not have provided with the hope of remission.
  • The initial symptoms for the bone relapse presented concurrently with the conditioning. The surgery had taken place after the patient had received 12 Gy total dose of irradiation. It was not possible to perform PET at the time of the neutropenic period. Moreover, several aspects of the presentation of the bone relapse indicated infectious osteomyelitis. It is not unsubstantiated, that concurrent infection and leukaemic invasion had taken place. It is not fully clear that the knowledge of PCR MRD positivity would have resulted in better guidance for such an ambiguous case. Later on, the potential multiplicity of the bone invasion was indeed considered. However, the extremely rapid relapse after the HSCT hindered these efforts of ours.

Comment about the immunophenotype:

For the IHC only B-cell markers were used, as the samples had partially deteriorated by that time. The correlation between the immunophenotype of the original and bone-invading blasts could not have been analysed because of this. This limitation is acknowledged at the end of the discussion part.

Comment about full therapy course with FC-MRD timepoints:

A figure has been added to the manuscript, that describes the main events of the case and the changes in flow-MRD during the course of the treatment.

Máté Horváth

Round 2

Reviewer 2 Report

The revised form of paper is clearer and better written. There are still some methodological problems, but they are not solvable.

Reviewer 3 Report

The authors responded appropriately to most queries. 

The significance is low /moderate but is ready for presentation.